# Mental health problems and social media exposure during COVID-19 outbreak

**Junling Gao**, **Pinpin Zheng, Yingnan Jia, Hao Chen, Yimeng Mao, Suhong Chen, Yi Wang, Hua Fu, Junming Dai***

School of Public Health, Fudan University, Fudan Institute of Health communication, Shanghai, China

* jmdai@fudan.edu.cn

**Data Availability Statement:** All relevant data are within the manuscript and its Supporting Information files.

**Funding:** Junling Gao was funded by National key R&D Program of China (grant no. 2018YFC2002000 & 2018YFC2002001) and

## Abstract

Huge citizens expose to social media during a novel coronavirus disease (COVID-19) outbroke in Wuhan, China. We assess the prevalence of mental health problems and examine their association with social media exposure. A cross-sectional study among Chinese citizens aged≥18 years old was conducted during Jan 31 to Feb 2, 2020. Online survey was used to do rapid assessment. Total of 4872 participants from 31 provinces and autonomous regions were involved in the current study. Besides demographics and social media exposure (SME), depression was assessed by The Chinese version of WHO-Five Well-Being Index (WHO-5) and anxiety was assessed by Chinese version of generalized anxiety disorder scale (GAD-7). multivariable logistic regressions were used to identify associations between social media exposure with mental health problems after controlling for covariates. The prevalence of depression, anxiety and combination of depression and anxiety (CDA) was 48.3% (95%CI: 46.9%-49.7%), 22.6% (95%CI: 21.4%-23.8%) and 19.4% (95%CI: 18.3%-20.6%) during COVID-19 outbroke in Wuhan, China. More than 80% (95%CI:80.9%-83.1%) of participants reported frequently exposed to social media. After controlling for covariates, frequently SME was positively associated with high odds of anxiety (OR = 1.72, 95%CI: 1.31–2.26) and CDA (OR = 1.91, 95%CI: 1.52–2.41) compared with less SME. Our findings show there are high prevalence of mental health problems, which positively associated with frequently SME during the COVID-19 outbreak. These findings implicated the government need pay more attention to mental health problems, especially depression and anxiety among general population and combating with "infodemic" while combating during public health emergency.

## Introduction

A public health emergency of international concern-novel coronavirus disease (COVID-19) outbroke[1] in Wuhan, China on 31 December 2019, which has been spread to 24 countries outside of China and infected 37,558 patients globally (37,251 in China) by 9 February 2020[2]. The outbreak of COVID-19 in China has caused mental health problems among the public in China[3] and Japan[4] and medical workers in Wuhan[5]. The National Health Commission

National Natural Science Foundation of China
(grant no. 71573048). The funders had no role in
study design, data collection and analysis, decision
to publish, or preparation of the manuscript.

has released guideline for local authorities to promote psychological crisis intervention for
patients, medical personnel, people under medical observation and civilians during the
COVID-19 outbreak[6]. However, what type of mental disorders are prevalent and how they
distribute among population are not know. So, a rapid assessment of outbreak-associated men-
tal disorders for both civilians and health care workers, is needed[7].

The official departments strive to improve the public's awareness of prevention and interven-
tion strategies by providing daily updates about surveillance and active cases on websites and
social media[3]. Besides, many self-media and netizens also release and transfer related infor-
mation on social media, such as WeChat and Weibo. Social media may lead to (mis)informa-
tion overload[8,9], which in turn may cause mental health problems. WHO pointed out that
identifying the underlying drivers of fear, anxiety and stigma that fuel misinformation and
rumour, particularly through social media[10]. Previous studies indicated that indirect exposure
to mass trauma through the media can increase the initial rates of post-traumatic stress disorder
(PTSD) symptoms[11]. A previous study also shown social media exposure may positively
related to forming risk perceptions during the MERS outbreak in South Korea[12]. But there
was no study to examine the association between social media exposure and mental health
problems. So, the current study aims to describes the prevalence and distribution of two major
mental disorders-anxiety and depression among Chinese population [13], and examine their
associations with social media exposure by rapid assessment during COVID-19 outbreak.

## Materials and methods

### Design and participants

This cross-sectional study was online conducted during Jan 31 to Feb 2, 2020. Chinese citizens
aged≥18 years old were invited to participate online survey though Wenjuanxing platform
(https://www.wjx.cn/app/survey.aspx). In total, 5,851 participants took part in the survey.
After removing the participants without completed questionnaires, 4872 participants from 31
provinces and autonomous regions were involved in the current study. A written consent in
the first section of online survey was given to all participants before filling the questionnaire.
This study has been approved by the Institutional Review Board of Fudan University, School
of Public Health(IRB#2020-01-0800).

### Measurements

**Mental health problems.** According to a previous study two major mental disorders-
depression and anxiety [13] were assessed in the current study. Depression was assessed by
The Chinese version of WHO-Five Well-Being Index (WHO-5)[14], which consists of five
positively worded items that reflect the presence or absence of well-being rather than depres-
sive symptomatology. Participants are asked to report the presence of these positive feelings in
the last 2 weeks on a 6-point scale ranging from all of the time (5 points) to at no time (0
points). A summed score below 13 indicates depression[14]. Anxiety was assessed by Chinese
version of generalized anxiety disorder scale (GAD-7)[15,16], which consists 7 symptoms. Par-
ticipants were asked how often they were bothered by each symptom during the last 2 weeks.
Response options were "not at all," "several days," "more than half the days," and "nearly every
day," scored as 0, 1, 2, and 3, respectively. A score of 10 or greater represents a reasonable cut
point for identifying cases of anxiety[15,16](S1 Table).

**Social media exposure (SME).** Social media exposure was measured by asking how often
respondents during the past week were exposed to news and information about COVID-19 on
social media, such as Sina weibo, Zhihu, Douban, WeChat and etc (S1 Table). Response
options were "never", "once in a while", "sometimes", "often" and "very often". Because of less

proportion of "never", so we recoded social media exposure into "less" ("never" and "once in a while"), "sometimes" and "frequently" ("often" and "very often").

**Covariates.** The following covariates were included in this study: gender, age (10-year categories), educational level (junior high school, senior high school, college and master and higher), marital status (recoded into married and other [including unmarried, divorced, and widowed]), self-rated health (categorized as excellent, very good and good or low), occupation(students/ retired, health care worker and others), cities(Wuhan and others), area(urban and rural).

## Statistical analyses

The $\chi^2$/trend tests were used to determine the prevalence of depression, anxiety and combination of depression and anxiety by categorical variables including social media exposure and covariates. Logistic regression analyses were used to explain the association between the prevalence of depression, anxiety and combination of depression and anxiety and SME after controlling for covariates. We estimated the adjusted ORs and their 95% confidence intervals (CIs) of independent variables for frailty. The STATA version 13.0 program (StataCorp LP., College Station, TX, USA) was used to carry out all analyses.

## Results

### Social media exposure

Of all 4827 participants, the mean age of was 32.3±10.0 years (ranged 18–85), the proportion of "less", "sometimes", and "frequently" of SME was 8.8%(95%CI:8.0%-9.6%), 9.2%(95% CI:8.4%-10.0%) and 82.0%(95%CI:80.9%-83.1%). As shown in **Table 1**, more than 60% of them (67.7%) were women, and most (47.9%) were aged 21–30 years. Many participants (62.2%) had achieved a college education, more than half of them were married. Only 5.2% of them were health care workers and 2.7% were from Hubei province, and 81.2% were from urban area. Most of them reported "excellent" (43.9%) or "very good" (45.6%) health.

Univariate analyses found that the proportion of frequently SME among men (78.4%, 95% CI: 76.3%-80.3%) was lower than among women (83.8%, 95%CI: 82.4%-85.0%), the proportion of frequently SME among youngers (aged -30 years) was higher than among elders (aged 41- years). Participants with low education (middle school and high school) had lower proportion of frequently SME than who with high education (college and master). Participants who are students or retired had higher proportion of frequently SME. The proportion of SME was not different between participants from Hubei province and others, however, participants from rural area reported higher proportion of frequently SME than who from urban area. Participants who were excellent health had higher proportion of frequently SME than others.

### Depression and SME

The prevalence of depression was 48.3% (95%CI: 46.9%-49.7%). As shown **Fig 1**, Multivariate analyses found that the adjusted odds of depression were greater among who age 21–30 years (OR = 1.49, 95%CI: 1.12–1.99) and 31–40 years (OR = 1.54, 95%CI: 1.11–2.14) compared with who aged ≤20 years, and lower among those with college (OR = 0.69, 95%CI: 0.53–0.91) and master (OR = 0.46, 95%CI: 0.63–0.85) education than those with middle school. Participants from Hubei province had no higher adjusted odds than those from other province (OR = 1.06, 95%CI: 0.75–1.52), but those from rural area had lower adjusted odds (OR = 0.74, 95%CI: 0.64–0.87) than those from urban area. The decrease of self-rated health significantly accompanied the increased odds of depression. About the focus of this study, higher frequency of SME was insignificantly positively associated with the adjusted odds of depression after controlling for all covariates.

**Table 1. Participants characteristic and social media exposure.**

| | N(%) | Social media exposure | | | P value |
|---|---|---|---|---|---|
| | | Less | Sometimes | Frequently | |
| **Overall** | 4827(100) | 424(8.8:8.0–9.6) | 444(9.2:8.4–10.0) | 3959(82.0:80.9–83.1) | |
| **Gender** | | | | | |
| Male | 1560(32.3) | 161(10.3:8.9–11.9) | 176(11.3:9.8–13.0) | 1223(78.4:76.3–80.3) | <0.001 |
| Female | 3267(67.7) | 263(8.1:7.2–9.0) | 268(8.2:7.3–9.2) | 2736(83.8:82.4–85.0) | |
| **Age(years)** | | | | | |
| -20 | 256(5.3) | 11(4.3:2.2–7.6) | 14(5.5:3.0–9.0) | 231(90.2:85.9–93.6) | <0.001 |
| 21–30 | 2312(47.9) | 120(5.2:4.3–6.2) | 150(6.5:5.5–7.6) | 2042(88.3:86.9–89.6) | |
| 31–40 | 1288(26.9) | 124(9.6:8.1–11.4) | 144(11.2:9.5–13.0) | 1020(79.2:76.9–81.4) | |
| 41–50 | 749(15.5) | 120(16.0:13.5–18.8) | 109(14.6:12.1–17.3) | 520(69.4:66.0–72.7) | |
| 51- | 222(4.6) | 49(22.1:16.8–28.1) | 27(12.2:8.2–17.2) | 146(65.8:59.1–72.0) | |
| **Education** | | | | | |
| Middle school | 257(5.3) | 45(17.5:13.1–22.7) | 30(11.7:8.0–16.2) | 182(70.8:64.8–76.3) | <0.001 |
| High School | 782(16.2) | 85(10.9:8.8–13.3) | 96(12.3:10.1–14.8) | 601(76.9:73.7–79.8) | |
| College | 3002(62.2) | 224(7.5:6.5–8.5) | 256(8.5:7.6–9.6) | 2522(84.0:82.6–85.3) | |
| Master | 786(16.3) | 70(8.9:7.0–11.1) | 62(7.9:6.1–10.0) | 654(83.2:80.4–85.8) | |
| **Marriage** | | | | | |
| Married | 2607(54.0) | 314(12.0:10.8–13.5) | 306(11.7:10.5–13.0) | 1987(76.2:74.5–77.8) | <0.001 |
| No married | 2220(46.0) | 110(5.0:4.1–5.9) | 138(6.2:5.2–7.3) | 1972(88.8:87.4–90.1) | |
| **Occupation** | | | | | |
| Students/retired | 1189(24.6) | 84(7.1:5.7–8.7) | 70(5.9:4.7–7.4) | 1035(87.1:85.0–88.9) | <0.001 |
| Health care workers | 251(5.2) | 25(10.0:6.5–14.4) | 23(9.2:5.9–13.4) | 203(80.9:75.5–85.6) | |
| others | 3387(70.2) | 315(9.3:8.3–10.3) | 351(10.4:9.4–11.4) | 2721(80.3:79.0–81.7) | |
| **Cities** | | | | | |
| Hubei | 130(2.7) | 7(5.4:2.2–10.8) | 13(10.0:5.4–16.5) | 110(84.6:77.2–90.3) | 0.375 |
| Others | 4697(97.3) | 417(8.9:8.1–9.7) | 431(9.2:8.4–10.0) | 3849(82.0:80.8–83.0) | |
| **Area** | | | | | |
| Urban | 3920(81.2) | 363(9.3:8.4–10.2) | 371(9.5:8.6–10.4) | 3186(81.3:80.0–82.5) | 0.015 |
| Rural | 907(18.8) | 61(6.7:5.2–8.6) | 73(8.1:6.4–10.0) | 773(85.2:82.7–87.5) | |
| **Self-rate health** | | | | | |
| Excellent | 2118(43.9) | 173(8.2:7.0–9.4) | 179(8.5:7.3–9.7) | 1766(83.4:81.7–84.9) | 0.020 |
| Very good | 2202(45.6) | 191(8.7:7.5–9.9) | 209(9.5:8.3–10.8) | 1802(81.8:80.2–83.4) | |
| Good/general/poor | 507(10.5) | 60(11.8:9.1–15.0) | 56(11.1:8.5–14.1) | 391(77.2:73.2–80.7) | |

## Anxiety and SME

The prevalence of anxiety was 22.6% (95%CI: 21.4%-23.8%). As shown **Fig 2**, Multivariate analyses found that that the adjusted odds of depression were greater among those aged 31–40 years (OR = 1.63, 95%CI: 1.06–2.51) compared with those aged -20 years, and lower among those with college (OR = 0.40, 95%CI: 0.30–0.53) and master (OR = 0.31, 95%CI: 0.22–0.44) education than those with middle school. The adjusted odds of depression among unmarried participants (OR = 0.80, 95%CI: 0.66–0.96) was lower than among married ones. Participants from other provinces had no higher adjusted odds (OR = 0.49, 95%CI: 0.33–0.71) than those from Hubei province. The adjusted odds of depression were greater among those with good/general/poor SRH (OR = 1.77, 95%CI: 1.41–2.21) compared with those with excellent SRH. About the focus of this study, frequently SME can increase the adjusted odds (OR = 1.72, 95% CI: 1.31–2.26) of anxiety compared with less SME after controlling for all covariates.

| Variables | Prevalence(95%CI) | | Adjusted odds ratio(95%CI) |
|---|---|---|---|
| **Overall** | **48.3(46.9-49.7)** | | |
| **Gender** | | | |
| Male | 46.9(44.4-49.4) | | 1(ref) |
| Female | 49.0(47.2-50.1) | | 0.99(0.87-1.12) |
| **Age(years)** | | | |
| -20 | 39.8(33.8-46.1) | | 1(ref) |
| 21-30 | 49.3(47.2-51.3) | | 1.49(1.12-1.99) |
| 31-40 | 49.8(47.0-52.5) | | 1.54(1.11-2.14) |
| 41-50 | 45.9(42.3-49.6) | | 1.21(0.85-1.71) |
| 51- | 47.3(40.6-50.1) | | 1.17(0.78-1.75) |
| **Education** | | | |
| Middle school | 51.8(45.5-58.0) | | 1(ref) |
| High School | 51.0(47.5-54.6) | | 0.88(0.65-1.17) |
| College | 47.7(45.9-49.5) | | 0.69(0.53-0.91) |
| Master | 46.6(43.0-50.1) | | 0.63(0.46-0.85) |
| **Marriage** | | | |
| Married | 48.1(46.1-50.0) | | 1(ref) |
| No married | 48.6(46.4-50.7) | | 1.07(0.91-1.26) |
| **Occupation** | | | |
| Students/retired | 46.2(43.3-49.1) | | 1(ref) |
| Health care workers | 49.4(43.1-55.8) | | 0.90(0.67-1.21) |
| others | 49.0(47.3-50.7) | | 0.97(0.74-1.27) |
| **Cities** | | | |
| Hubei | 47.7(38.9-56.6) | | 1(ref) |
| Others | 48.3(46.9-49.7) | | 1.06(0.75-1.52) |
| **Area** | | | |
| Urban | 49.4(47.8-51.0) | | 1(ref) |
| Rural | 43.4(40.2-46.7) | | 0.74(0.64-0.87) |
| **Self-rate health** | | | |
| Excellent | 40.2(38.1-42.4) | | 1(ref) |
| Very good | 52.2(50.1-54.3) | | 1.69(1.49-1.91) |
| Good/general/poor | 65.1(60.8-69.2) | | 2.91(2.37-3.58) |
| **Social media exposure** | | | |
| Less | 45.8(40.9-50.6) | | 1(ref) |
| Sometimes | 47.5(42.8-52.3) | | 1.07(0.81-1.41) |
| Frequently | 48.6(47.1-50.2) | | 1.18(0.96-1.45) |

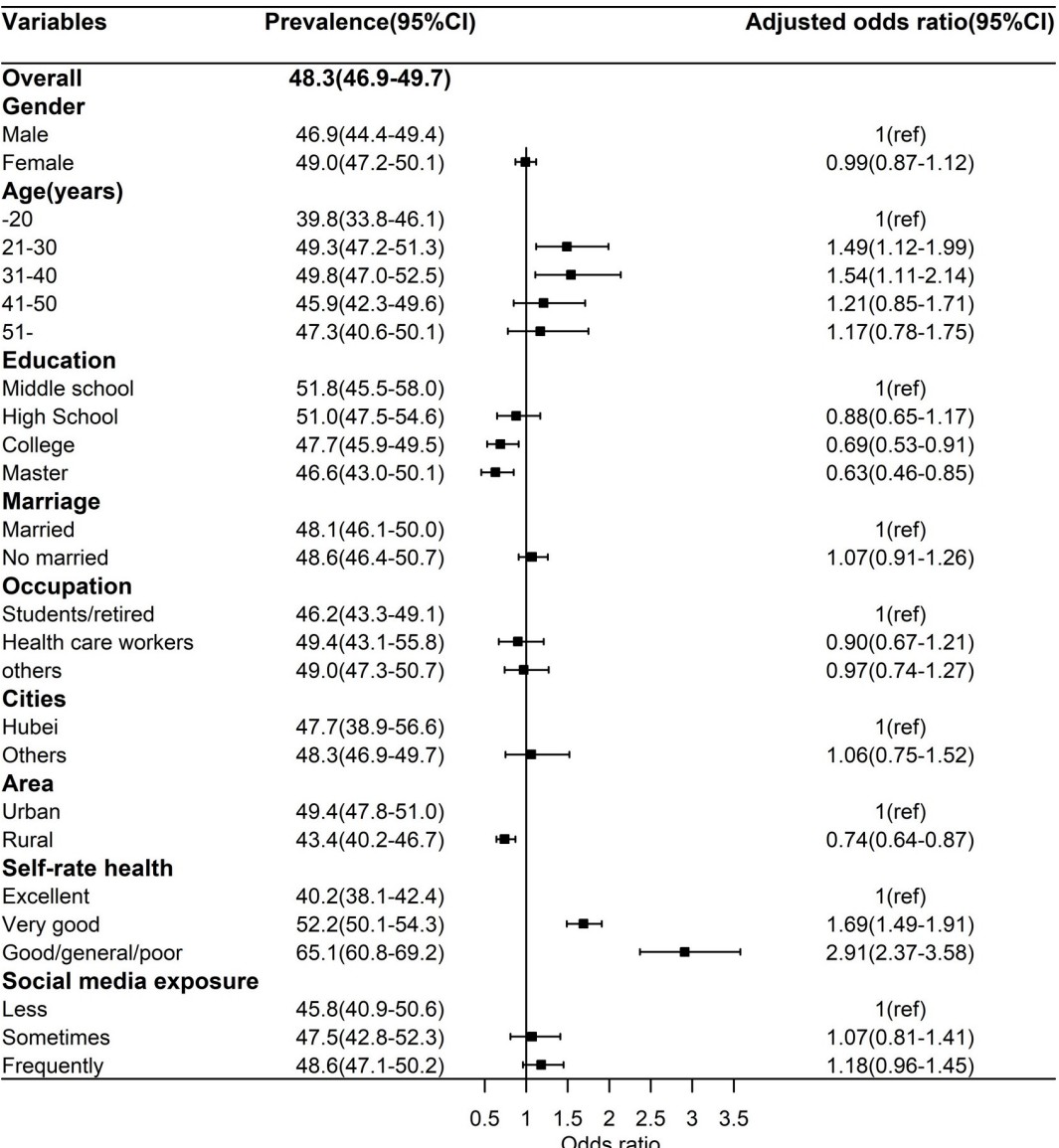

**Fig 1. Prevalence of depression and relevant factors.**

## Combination of depression and anxiety and SME

The prevalence of combination of depression and anxiety (CDA) was 19.4% (95%CI: 18.3%-20.6%). As shown **Fig 3**, Multivariate analyses found that that the adjusted odds of depression were greater among those aged 31–40 years (OR = 1.69, 95%CI: 1.07–2.68) compared with those aged -20 years, and lower among those with college (OR = 0.50, 95%CI: 0.37–0.68) and master (OR = 0.40, 95%CI: 0.28–0.57) education than those with middle school. The adjusted odds of depression among unmarried participants (OR = 0.79, 95%CI: 0.64–0.97) was lower than among married ones. The adjusted odds of depression were greater among those with good/general/poor SRH (OR = 1.77, 95%CI: 1.41–2.21) compared with those with excellent SRH. About the focus of this study, frequently SME can increase the adjusted odds (OR = 1.91, 95%CI: 1.52–2.41) of CDA compared with less SEM after controlling for all covariates.

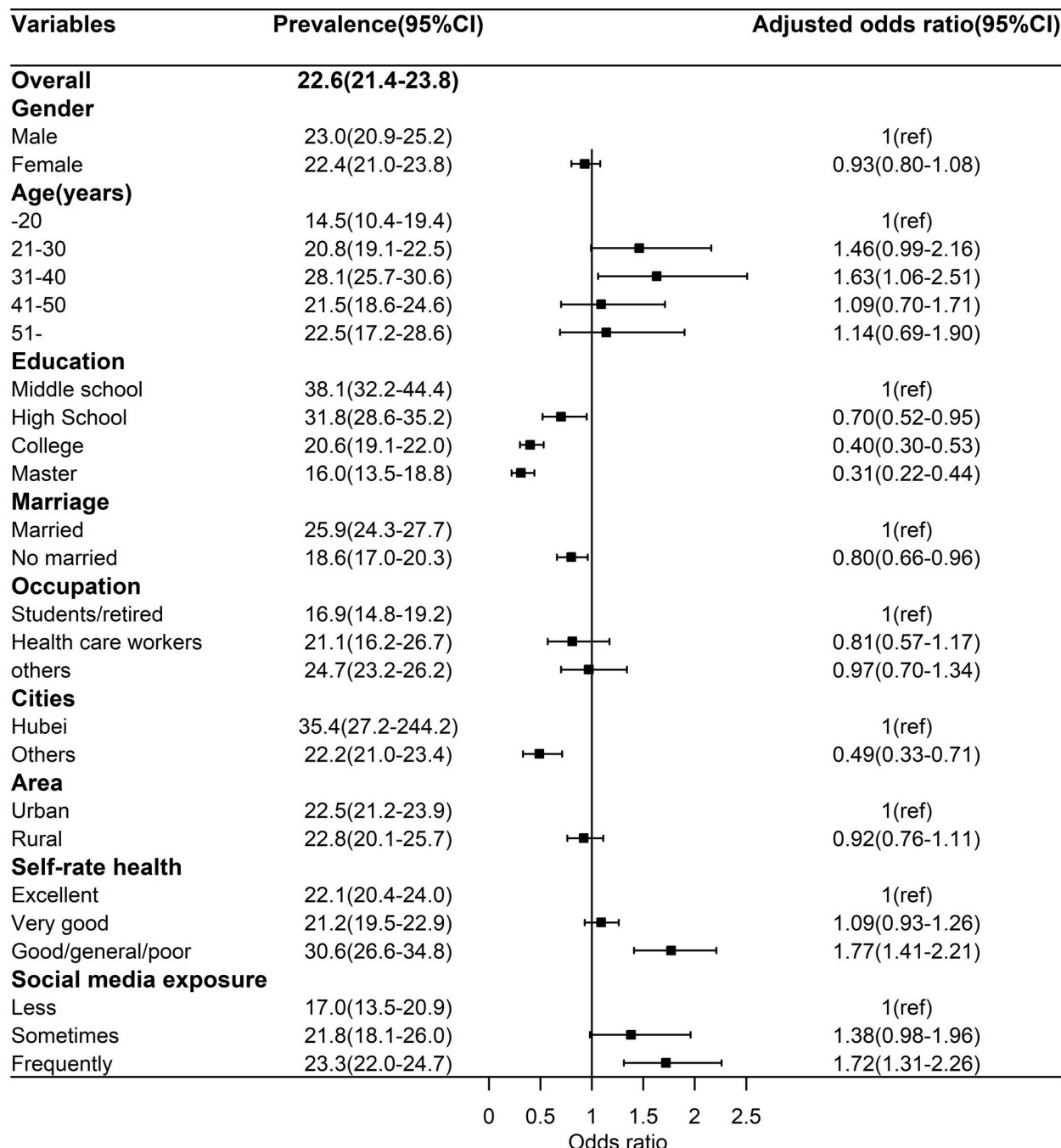

| Variables | Prevalence(95%CI) | Adjusted odds ratio(95%CI) |
|---|---|---|
| **Overall** | **22.6(21.4-23.8)** | |
| **Gender** | | |
| Male | 23.0(20.9-25.2) | 1(ref) |
| Female | 22.4(21.0-23.8) | 0.93(0.80-1.08) |
| **Age(years)** | | |
| -20 | 14.5(10.4-19.4) | 1(ref) |
| 21-30 | 20.8(19.1-22.5) | 1.46(0.99-2.16) |
| 31-40 | 28.1(25.7-30.6) | 1.63(1.06-2.51) |
| 41-50 | 21.5(18.6-24.6) | 1.09(0.70-1.71) |
| 51- | 22.5(17.2-28.6) | 1.14(0.69-1.90) |
| **Education** | | |
| Middle school | 38.1(32.2-44.4) | 1(ref) |
| High School | 31.8(28.6-35.2) | 0.70(0.52-0.95) |
| College | 20.6(19.1-22.0) | 0.40(0.30-0.53) |
| Master | 16.0(13.5-18.8) | 0.31(0.22-0.44) |
| **Marriage** | | |
| Married | 25.9(24.3-27.7) | 1(ref) |
| No married | 18.6(17.0-20.3) | 0.80(0.66-0.96) |
| **Occupation** | | |
| Students/retired | 16.9(14.8-19.2) | 1(ref) |
| Health care workers | 21.1(16.2-26.7) | 0.81(0.57-1.17) |
| others | 24.7(23.2-26.2) | 0.97(0.70-1.34) |
| **Cities** | | |
| Hubei | 35.4(27.2-244.2) | 1(ref) |
| Others | 22.2(21.0-23.4) | 0.49(0.33-0.71) |
| **Area** | | |
| Urban | 22.5(21.2-23.9) | 1(ref) |
| Rural | 22.8(20.1-25.7) | 0.92(0.76-1.11) |
| **Self-rate health** | | |
| Excellent | 22.1(20.4-24.0) | 1(ref) |
| Very good | 21.2(19.5-22.9) | 1.09(0.93-1.26) |
| Good/general/poor | 30.6(26.6-34.8) | 1.77(1.41-2.21) |
| **Social media exposure** | | |
| Less | 17.0(13.5-20.9) | 1(ref) |
| Sometimes | 21.8(18.1-26.0) | 1.38(0.98-1.96) |
| Frequently | 23.3(22.0-24.7) | 1.72(1.31-2.26) |

**Fig 2. Prevalence of anxiety and relevant factors.**

## Discussion

The latest national sample indicated the prevalence of any disorder (excluding dementia), anxiety disorders and depressive disorders was 16.6% (95%CI: 13.0–20.2), 7.6% (95%CI: 6.3–8.8) and 6.9% (95%CI: 6.6–7.2) in China[13]. Comparing with this national data, the current cross-sectional study found that much higher prevalence of depression (48.3%, 95%CI: 46.9%-49.7%), anxiety (22.6%, 95%CI: 21.4%-23.8%) and CDA (19.4%, 95%CI: 18.3%-20.6%) during

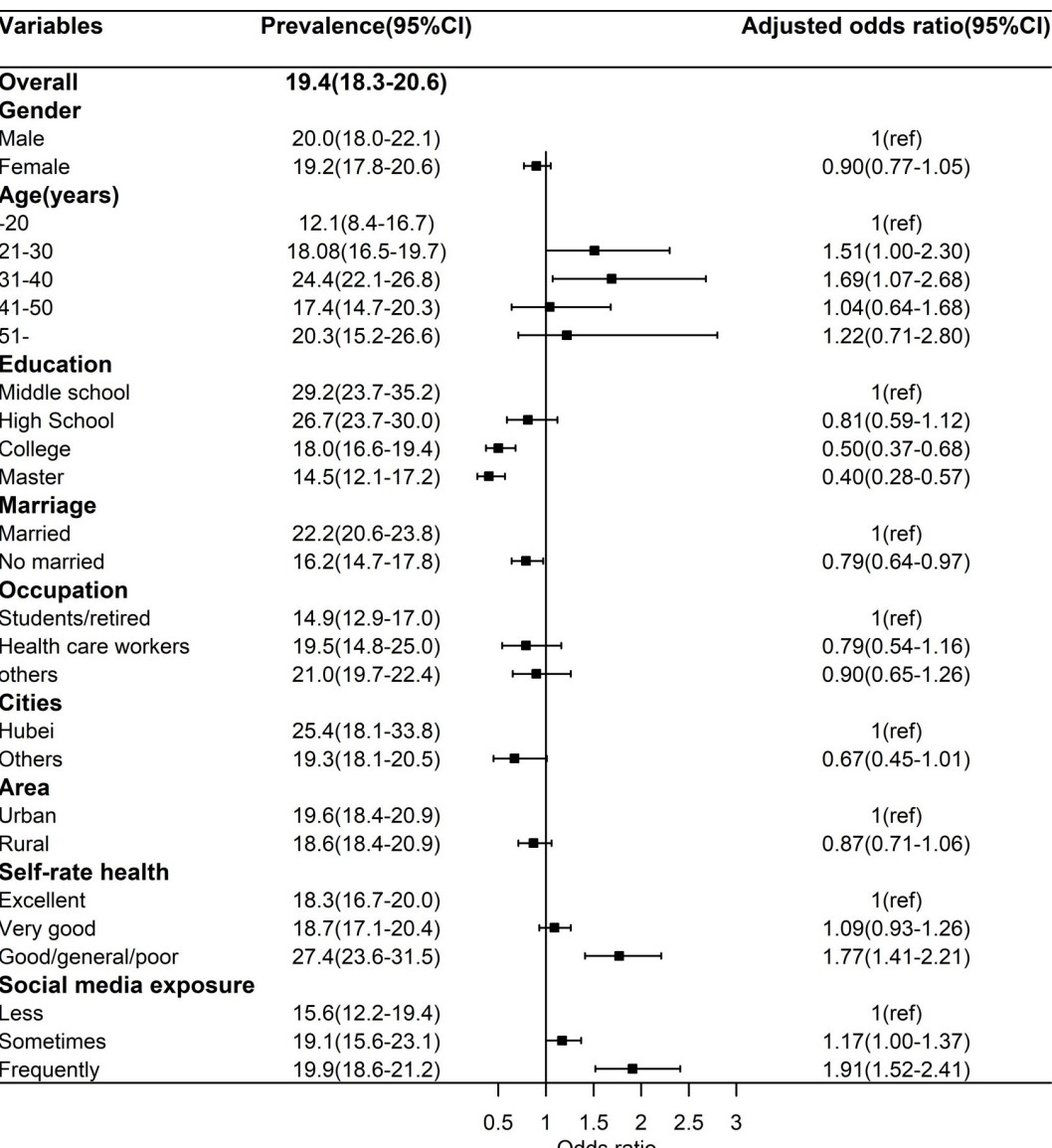

| Variables | Prevalence(95%CI) | Adjusted odds ratio(95%CI) |
|---|---|---|
| **Overall** | 19.4(18.3-20.6) | |
| **Gender** | | |
| Male | 20.0(18.0-22.1) | 1(ref) |
| Female | 19.2(17.8-20.6) | 0.90(0.77-1.05) |
| **Age(years)** | | |
| -20 | 12.1(8.4-16.7) | 1(ref) |
| 21-30 | 18.08(16.5-19.7) | 1.51(1.00-2.30) |
| 31-40 | 24.4(22.1-26.8) | 1.69(1.07-2.68) |
| 41-50 | 17.4(14.7-20.3) | 1.04(0.64-1.68) |
| 51- | 20.3(15.2-26.6) | 1.22(0.71-2.80) |
| **Education** | | |
| Middle school | 29.2(23.7-35.2) | 1(ref) |
| High School | 26.7(23.7-30.0) | 0.81(0.59-1.12) |
| College | 18.0(16.6-19.4) | 0.50(0.37-0.68) |
| Master | 14.5(12.1-17.2) | 0.40(0.28-0.57) |
| **Marriage** | | |
| Married | 22.2(20.6-23.8) | 1(ref) |
| No married | 16.2(14.7-17.8) | 0.79(0.64-0.97) |
| **Occupation** | | |
| Students/retired | 14.9(12.9-17.0) | 1(ref) |
| Health care workers | 19.5(14.8-25.0) | 0.79(0.54-1.16) |
| others | 21.0(19.7-22.4) | 0.90(0.65-1.26) |
| **Cities** | | |
| Hubei | 25.4(18.1-33.8) | 1(ref) |
| Others | 19.3(18.1-20.5) | 0.67(0.45-1.01) |
| **Area** | | |
| Urban | 19.6(18.4-20.9) | 1(ref) |
| Rural | 18.6(18.4-20.9) | 0.87(0.71-1.06) |
| **Self-rate health** | | |
| Excellent | 18.3(16.7-20.0) | 1(ref) |
| Very good | 18.7(17.1-20.4) | 1.09(0.93-1.26) |
| Good/general/poor | 27.4(23.6-31.5) | 1.77(1.41-2.21) |
| **Social media exposure** | | |
| Less | 15.6(12.2-19.4) | 1(ref) |
| Sometimes | 19.1(15.6-23.1) | 1.17(1.00-1.37) |
| Frequently | 19.9(18.6-21.2) | 1.91(1.52-2.41) |

**Fig 3. Prevalence of combination of depression and anxiety and relevant factors.**

COVID-19 outbroke in Wuhan, China. These findings are consistent with the previous studies' that exposing public health emergency can cause public mental health problems, such as Wenchuan and Lushan earthquakes[17], 2014 Ebola Outbreak[7,18], and SARS[19].

Social media is one of main channels updating the COVID-19 information[3]. This study also found that 82.0% of participants frequently expose them to social media, and frequently SME associated high odds of anxiety and CDA, which is consistent with previous studies [11]. there may be two reasons explaining the association between frequently SME and mental health. During COVID-19 outbreak, disinformation and false reports about the COVID-19 have bombarded social media and stoked unfounded fears among many netizens[20], which may confuse people and harm people's mental health[9]. Besides, many citizens expressed their negative feelings, such as fear, worry, nervous, anxiety et al. on social media, which are contagious social network[21,22]. So, WHO's 'infodemics' team is working hand in glove with countries'

communications department to deliver information to a broader public audience[23]. Finally, we also found that SME was not different between participants from Hubei province and others, but the formers faced higher odds of anxiety. It indicated that participants from Hubei province- the infectious focus directly expose to public health emergency, and may suffer more mental health problemes[17,19]. Compared with the control measures taken by other cities, Wuhan have sealed off the city from all outside contact to stop the spread of the COVID-19. As the prevention and control measures called new standard by WHO[24], the lockdown of Wuhan is a very effective way to interrupt the transmission of the virus, however, the strictest measures in Wuhan might lead to more serious mental health problems of local people.

Some potential limitations should be noted in this study. First, this is a cross-sectional study, so it is difficult to accurately elucidate causal relationships between SME and mental health. Additional longitudinal studies, such as cohort studies or nested case-control studies, are essential in the future. Although large sample, the survey was conducted online, which is suitable for rapid assessment, so some respondent bias, such as few elder citizens' participation, may have affected the results. Finally, although we did control for many covariates, we cannot exclude the possibility of some residual confounding caused by unmeasured factors.

## Conclusions

In conclusion, our findings show there are high prevalence of mental health problems, which positively associated with frequently SME during the COVID-19 outbreak. These findings implicated the government need pay more attention to mental health among general population while combating with COVID-19. Fortunately, The China government have provided mental health services by varied channel including hotline, online consultation, online course and outpatient consultation[6], but more attention should be paid to depression and anxiety. The next implication is to combat with "infodemic" by monitoring and filtering out false information and promoting accurate information though cross-section collaborations.

## Supporting information

**S1 Table.**
(DOCX)

**S1 Data.**
(XLS)

## Author Contributions

**Conceptualization:** Junling Gao, Hua Fu, Junming Dai.

**Data curation:** Pinpin Zheng, Yingnan Jia, Hao Chen, Yimeng Mao, Suhong Chen, Yi Wang.

**Formal analysis:** Junling Gao.

**Funding acquisition:** Junling Gao.

**Investigation:** Pinpin Zheng, Yingnan Jia, Hao Chen, Yimeng Mao, Suhong Chen, Yi Wang.

**Methodology:** Junling Gao, Yingnan Jia, Junming Dai.

**Supervision:** Hua Fu.

**Writing – original draft:** Junling Gao.

**Writing – review & editing:** Pinpin Zheng, Yingnan Jia, Hua Fu, Junming Dai.

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
