## [Decision Letter · Decision Letter 0]

1 Apr 2020

PONE-D-20-06332

Mental health problems and social media exposure during COVID-19 outbreak

PLOS ONE

Dear Dr. Gao,

Thank you for submitting your manuscript to PLOS ONE. After careful consideration, we feel that it has merit but does not fully meet PLOS ONE’s publication criteria as it currently stands. Therefore, we invite you to submit a revised version of the manuscript that addresses the points raised during the review process.

Your manuscript is timely and interesting for the readers of the journal. But the reviewer #1 addressed several major concerns about your manuscript. Please revise your manuscript carefully and ASAP.

We would appreciate receiving your revised manuscript by May 16 2020 11:59PM. To enhance the reproducibility of your results, we recommend that if applicable you deposit your laboratory protocols in protocols.io, where a protocol can be assigned its own identifier (DOI) such that it can be cited independently in the future. For instructions see: http://journals.plos.org/plosone/s/submission-guidelines#loc-laboratory-protocols

We look forward to receiving your revised manuscript.

Kind regards,

Kenji Hashimoto, PhD

Academic Editor

PLOS ONE

Journal Requirements:

2. Please provide additional details regarding participant consent. In the ethics statement in the Methods and online submission information, please ensure that you have specified (1) whether consent was suitably informed and (2) what type you obtained (for instance, written or verbal). If the need for consent was waived by the ethics committee, please include this information.

3. Please include additional information regarding the survey or questionnaire used in the study and ensure that you have provided sufficient details that others could replicate the analyses. If you developed and/or translated a questionnaire as part of this study and it is not under a copyright license more restrictive than Creative Commons Attribution (CC-BY), please include a copy, in both the original language and English, as Supporting Information.

5. Please include your tables as part of your main manuscript and remove the individual files. Please note that supplementary tables (should remain/ be uploaded) as separate "supporting information" files

Additional Editor Comments (if provided):

Reviewers' comments:

Reviewer's Responses to Questions

**Comments to the Author**

1. Is the manuscript technically sound, and do the data support the conclusions?

Reviewer #1: Partly

Reviewer #2: Yes

2. Has the statistical analysis been performed appropriately and rigorously? 

Reviewer #1: Yes

Reviewer #2: Yes

3. Have the authors made all data underlying the findings in their manuscript fully available?

Reviewer #1: Yes

Reviewer #2: Yes

4. Is the manuscript presented in an intelligible fashion and written in standard English?

Reviewer #1: No

Reviewer #2: Yes

5. Review Comments to the Author

Reviewer #1: 1、In the result sections “Anxiety and SME” and “Combination of depression and Anxiety and SME”, the author also used the keyword “depression”.

2、The authors revealed the positive relationship between SME and anxiety, however, the value of this finding has not been described clearly and the advice for application is not rational. Because of the lack of causal evidence, we cannot know whether the higher SME is the reason or the result of higher anxiety. There seems no convincing reason to leave both the general suggestion of combating with “infodemic”, as well as the specific suggestions against false information and rumors, since there is no data at all in this article to distinguish the influence from true or false social media information on those cases with anxiety.

If the authors could provide more data on WHY frequent SME associated with high odds of anxiety and CDA, their implications about infodemic would be more reasonable.

Reviewer #2: The authors investigated the relationship between social media exposure and the prevalence of mental health problems during a novel coronavirus disease (COVID-19) outbreak in Wuhan, China. They found that there are high prevalence of

mental health problems, which positively associated with frequently social media exposure during this infectious outbreak.

The subject of this manuscript is important and interesting in the present world situation, and the findings may be helpful for the design of future infectious disease outbreak management. I recommend this for the publication in this journal.

6. PLOS authors have the option to publish the peer review history of their article (what does this mean?). If published, this will include your full peer review and any attached files.

Reviewer #1: No

Reviewer #2: No

---

## [Author Response · Author response to Decision Letter 0]

2 Apr 2020

Dear editor and reviewers,

We would like to thank you for your constructive comments and suggestions. We have revised the manuscript accordingly and would like to resubmit it for your consideration. Below, we have outlined our responses to each of the comments provided by the academic editor and reviewers. 

Thank you for your consideration. We look forward to hearing from you soon.

Yours sincerely,

Junling Gao

Journal Requirements:

Response: the manuscript’s style was revised according to the journal style requirements.

2. Please provide additional details regarding participant consent. In the ethics statement in the Methods and online submission information, please ensure that you have specified (1) whether consent was suitably informed and (2) what type you obtained (for instance, written or verbal). If the need for consent was waived by the ethics committee, please include this information.

Response: a written consent was given to every participant before filling the questionnaire. The following sentences were revised in the manuscript.

A written consent in the first section of online survey was given to all participants before filling the questionnaire. This study has been approved by the Institutional Review Board of Fudan University, School of Public Health(IRB#2020-01-0800). 

3. Please include additional information regarding the survey or questionnaire used in the study and ensure that you have provided sufficient details that others could replicate the analyses. If you developed and/or translated a questionnaire as part of this study and it is not under a copyright license more restrictive than Creative Commons Attribution (CC-BY), please include a copy, in both the original language and English, as Supporting Information.

Response: the questionnaires were used to measure SME and depression and anxiety were provided as Supporting Information.

4. We note that you have indicated that data from this study are available upon request. PLOS only allows data to be available upon request if there are legal or ethical restrictions on sharing data publicly.

Response: the minimal anonymized data set was shared

5. Please include your tables as part of your main manuscript and remove the individual files.

Response: table was added in the revised manuscript.

Reviewers’ comments

Reviewer #1: 

1、In the result sections “Anxiety and SME” and “Combination of depression and Anxiety and SME” , the author also used the keyword “depression”.

Response: the keywords and titles were revised according to the journal requirements

2、The authors revealed the positive relationship between SME and anxiety, however, the value of this finding has not been described clearly and the advice for application is not rational. Because of the lack of causal evidence, we cannot know whether the higher SME is the reason or the result of higher anxiety. There seems no convincing reason to leave both the general suggestion of combating with “infodemic”, as well as the specific suggestions against false information and rumors, since there is no data at all in this article to distinguish the influence from true or false social media information on those cases with anxiety.

If the authors could provide more data on WHY frequent SME associated with high odds of anxiety and CDA, their implications about infodemic would be more reasonable.

Response: thank you for your crucial suggestion. Because this is a cross-sectional study, we can elucidate causal relationships between SME and mental health, which has been mentioned in the limitation section.

Based on literature, we think there may be two reasons as following

 During COVID-19 outbreak, disinformation and false reports about the COVID-19 have bombarded social media and stoked unfounded fears among many netizens[20], which may confuse people and harm people’s mental health[9]. This study found that frequently SME associated high odds of anxiety and CDA, which is consistent with previous studies.11 Besides, many citizens expressed their negative feelings, such as fear, worry, nervous, anxiety et al. on social media, which are contagious social network[21,22].

As for this limitation, we revised the implications in order to make them consisting with results according to your suggestions.

Reviewer #2: The authors investigated the relationship between social media exposure and the prevalence of mental health problems during a novel coronavirus disease (COVID-19) outbreak in Wuhan, China. They found that there are high prevalence of

mental health problems, which positively associated with frequently social media exposure during this infectious outbreak.

The subject of this manuscript is important and interesting in the present world situation, and the findings may be helpful for the design of future infectious disease outbreak management. I recommend this for the publication in this journal.

Response: thank you for your comments.

---

## [Editor Report · Decision Letter 1]

6 Apr 2020

Mental health problems and social media exposure during COVID-19 outbreak

PONE-D-20-06332R1

Dear Dr. Gao,

We are pleased to inform you that your manuscript has been judged scientifically suitable for publication and will be formally accepted for publication once it complies with all outstanding technical requirements.

With kind regards,

Kenji Hashimoto, PhD

Section Editor

PLOS ONE
---

## [Editor Report · Acceptance letter]

9 Apr 2020

PONE-D-20-06332R1 

Mental health problems and social media exposure during COVID-19 outbreak 

Dear Dr. Gao:

I am pleased to inform you that your manuscript has been deemed suitable for publication in PLOS ONE. Congratulations! Your manuscript is now with our production department. 

With kind regards,

on behalf of

Prof. Kenji Hashimoto 

Section Editor

PLOS ONE